# AltDev: Achieving Real-Time Alignment in Multi-Agent Software Development

## Abstract

Large Language Models (LLMs) have shown remarkable capability in code generation tasks. However, they still struggle with complex software development tasks where agents of different roles need to work collaboratively. Existing works have proposed some LLM-based multi-agent software development frameworks following linear models such as the Waterfall model. However, linear models suffer from erroneous outputs of LLMs due to the lack of a self-correction mechanism. Inspired by human teams where people can freely start meetings for reaching agreement, we propose a novel and flexible multi-agent framework AltDev, which enables agents to correct their deliverables and align with other agents in a real-time manner. AltDev integrates a compulsory alignment checking and a conditional multi-agent discussion at the end of each development phase, in order to identify and reduce errors at early stages in the software development lifecycle. Our experiments on various software development tasks show that AltDev significantly improves the quality of generated software code in terms of executability, structural and functional completeness. The code of our project is available at https://github.com/RainingSea/Altdev.

## 1 Introduction

Large Language Models (LLMs) have been demonstrated effective in various code generation tasks and become efficient assistants to human developers (Austin et al., 2021; Chen et al., 2021b). However, single LLM performs poor when facing complex software development projects, where the internal logics of functionalities could be super complicated. A natural extension is to incorporate multiple LLM-based roles in the software development team, including product managers, architects, project managers, programmers and test engineers. Due to the fast speed of content generation and communication, LLM-based software development teams can be much more efficient than human teams, making it a promising approach for automated software development.

Despite the efficiency of LLMs in software development, the misalignment between LLM agents becomes a significant challenge. Actually, the misalignment is also very common in real-world software development. For example, the architect might ignore some requirements and draw an incomplete architecture diagram. Also the programmer might misunderstood some functionalities in the architecture diagram and produce erroneous codes. Figure 1 gives a concrete example to illustrate the misalignment. Such misalignment is even worse in LLM-based development teams due to the hallucination of LLMs. Moreover, as the number of agents increases, the misalignment between agents will accumulate and finally lead to the failure of the whole project. Therefore, it is critical to develop mechanisms to reduce such misalignment before applying LLM-based agents to real-world complex software projects.

In order to facilitate software development, people have developed many mature models through decades of practice, including the Waterfall model, V-model and Agile model (Sundramoorthy & Murugaiyan, 2012; Adetokunbo & Adenowo, 2013). These models rely heavily on the intelligence and autonomy of humans, thus cannot directly apply to LLM-based agents. Recent works have proposed several workflows for LLM-based agents following the Waterfall model (Adetokunbo & Adenowo, 2013). For example, MetaGPT encodes Standardized Operating Procedures (SOPs) into prompt sequences for a streamlined workflow. However, they only test code quality in the end of the whole procedures, without an effective self-correction mechanism during the development pro-

Figure 1: An example illustrating the misalignment during the software development process. The original requirement is a sticker that can be zoom in and zoom out by users. Unfortunately, the project manager misunderstands the requirement and writes a misleading function description (shown in red). Consequently, the programmer assigns a fixed size to the sticker in her codes.

cess. ChatDev introduces a chat-powered software development framework, where several assistant agents are employed for code reviews (Hong et al., 2024). Unfortunately, the agents in ChatDev can only communicate through dialogues, therefore fail to check the correctness of intermediate deliverables such as architect diagrams.

The main drawback of the plain Waterfall model is that it only test the final codes at the last step. As a consequence, if any errors occurred during the development process, the whole project should start over. To overcome this issue, we propose a novel LLM-based multi-agent framework AltDev, which aims to enhance the quality management of software development during the whole development process. Following existing works (Adetokunbo & Adenowo, 2013; Hong et al., 2024), AltDev employs five roles played by LLM-based agents, who work collaboratively to develop a certain software. Specifically, given a description of requirements, the *product manager* outputs a detailed and numerated requirement document. Then, the *architect* translates the requirement document to an architect diagram. Hence, a *project manager* sets out a task plan for the *programmer* to execute. Finally, a *test engineer* will test if the program meets the original requirements.

To address the misalignment during the development process, AltDev introduces an effective real-time alignment mechanism, inspired by human development teams where people could start meetings anytime to reach an agreement. Specifically, AltDev maintains a Shared Certified Repository (SCR) to store the certified contents, including requirement documents, architecture diagrams, task plans and codes. At the end of each phase in the workflow, the generated contents will go through an alignment checking procedure before being added to the SCR. Whenever a misalignment is detected, the agent responsible for the current phase would initialize a discussion to address the misalignment, and then regenerate contents based on the discussion. The above process will repeat until the generated contents pass the alignment checking. Note that AltDev actually integrates the Waterfall model with a non-linear self-correction mechanism. By this way, AltDev is more robust to erroneous outputs of LLMs and potentially leads to higher success rate of the project.

Our contributions can be summarized as follows:

• We introduce a novel LLM-based multi-agent software development framework AltDev. Compared with existing linear models, AltDev introduces a non-linear real-time alignment mechanism for correcting erroneous outputs, thus can can solve more complex software development tasks.

• We propose a Chain-of-Thought (CoT) inspired prompting method called Chain-of-Checking (CoC) for alignment checking at the end of each development phase. CoC efficiently guides different roles of LLM-based agents to check if the current deliverables align with previous ones.

• We present a set of standardized prompt sequences to implement AltDev. The library of prompt sequences is provided in Appendix A.2 for reproduction of AltDev.

• We test AltDev on over 200 real-world software development tasks from various domains. Experimental results show that AltDev largely improves the quality of generated codes in terms of executability, structural and functional completeness. Moreover, under the framework of AltDev, the misalignment between agents can be significantly reduced compared with baselines.

## 2 RELATED WORKS

### 2.1 CLASSIC SOFTWARE DEVELOPMENT MODELS

Through decades of practice, people have developed many mature software development models regarding to various scenarios. For example, the Waterfall model is a sequential and linear software development process that breaks down the software development lifecycle into distinct, discrete phases (Adetokunbo & Adenowo, 2013). The V-model is an evolution of the traditional Waterfall model, which introduces parallelism and feedback loops between development and testing activities (Sundramoorthy & Murugaiyan, 2012). The Agile model is a flexible and iterative approach that emphasizes rapid delivery of high-quality software through close collaboration and continuous improvement (Samar et al., 2020). However, these models only describe general workflows and principles, whose implementations depend heavily on the interpretation of human experts. Therefore, these models cannot directly apply to LLM-based development teams.

### 2.2 LLMs FOR CODE GENERATION

Large Language Models (LLMs) have shown surprising performance across a wide range of natural language processing tasks (Radford et al., 2019; Ouyang et al., 2022; Achiam et al., 2023; Touvron et al., 2023). LLMs also significantly boost the performance of code generation tasks and demonstrate a promising potential to be applied at scale. For example, Codex achieves a success rate of 28.8% in solving a set of 164 hand-written programming problems (Chen et al., 2021a). Copilot, a code generation tool powered by Codex, has captured the interest of over 1 million professional developers (Microsoft, 2023). Other models, including Incoder (Fried et al., 2023), CodeRL (Lea et al., 2022), Code Llama (Roziere et al., 2023) and ChatGPT (OpenAI, 2022), also achieve human-level performance in various code generation tasks. However, these works focus only on code generation ability of single LLMs, ignoring the collaboration and communication between multiple LLMs, which is crucial to complex software development tasks.

### 2.3 MULTI-AGENT FRAMEWORKS FOR SOFTWARE DEVELOPMENT

The key challenge in multi-agent software development is to efficiently coordinate with diverse roles of agents while ensuring consistent understanding of the outcomes (Unterkalmsteiner et al., 2015; Bjarnason et al., 2019). Several mutli-LLM software development frameworks have been proposed to enhance collaboration between agents. For example, the Self-Collaboration framework assigns different roles to LLMs and encourage them to finish sub-tasks collaboratively (Dong et al., 2024). MetaGPT proposes the first concrete and standardized software development framework, where multiple LLM-based agents complete different tasks following a linear workflow (Hong et al., 2024). However, MetaGPT fails ensure the alignment of agents' understandings about the current project, thus performs poor in complex projects. ChatDev introduces a set of assistant agents to verify if the generated codes are compliant with original requirements (Qian et al., 2023). However, agents in ChatDev communicate through dialogues, thus they can only ensure the alignment between requirements and codes. By contrast, the communication mechanism in our framework supports multiple types of contents, including requirement documents, architecture diagrams, task plans, codes and reviews, which enables AltDev to achieve alignment in the whole development process.

## 3 METHODOLOGY

In this section, we present the details of our framework AltDev. Generally, AltDev consists of a general workflow and a real-time alignment mechanism, which consists of a compulsory alignment checking phase and a conditional multi-agent discussion phase.

### 3.1 GENERAL WORKFLOW

Our general workflow is extended from the Waterfall model, which is originated from human software development (Adetokunbo & Adenowo, 2013). The Waterfall model specifies several important roles in software development and force them to work in a linear manner. Due to its simplicity, Waterfall model and its variants has been employed in some LLM-based software development

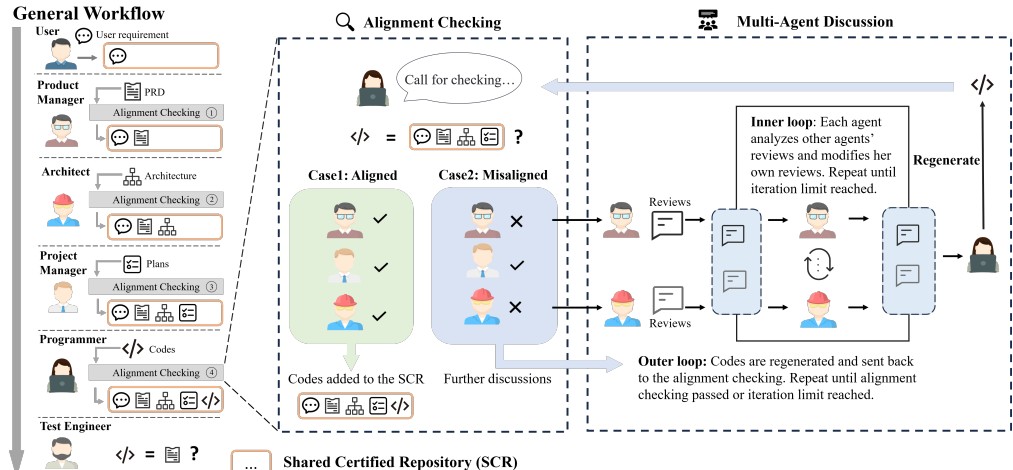

Figure 2: The overall framework of AltDev. The left part shows the general workflow, where five LLM-based agents work in a Waterfall manner to accomplish requirements inputted by the user. The middle part shows the alignment checking phase, in which all previous agents need to check if the newly generated contents are consistent with outcomes stored in a Shared Certified Repository. The right part in a conditional multi-agent discussion phase, depending on whether the newly generated contents pass the alignment checking. After discussion, the current agent should regenerate contents for another round of alignment checking. The loop will stop until the regenerated contents pass the alignment checking or the maximum number of iterations is reached. We only showcase the alignment checking and the multi-agent discussion for the programmer agent due to space limit.

frameworks. For example, the Standardized Operating Procedures encoded by MetaGPT is essentially a Waterfall model. ChatDev proposes a chat chain with sequential phases following the principles of the Waterfall model. The main drawback of the Waterfall model is that is does not allow changes to intermediate deliverables due to its linear nature. Our work extends the Waterfall model by incorporating additional alignment checking and multi-agent discussion procedures, so that erroneous outputs can be corrected in real-time. In fact, our framework is similar to the V-model in software engineering, where several review processes are enforced from the code level to the requirement level (Sundramoorthy & Murugaiyan, 2012). Unlike V-model where the review processes start at the end of the whole development process, AltDev is more flexible and efficient since it reduces any misalignment at its occurrence during the development process. As shown in Figure 2, the framework of AltDev includes the following five roles of agents played by LLMs. Note that each role is played by a single LLM agent in this work. But our framework can be extended to involve more agents by further task decomposition.

**Product Manager.** The Product Manager agent performs requirement analysis based on the raw functional description of some target software (Arora et al., 2024; Jin et al., 2024). In our framework, we ask the Product Manager agent to output the Product Requirement Document (PRD) as a list of numerated requirements that describe the functionalities of the target software.

**Architect.** The Architect agent analyzes the PRD and determines the general architecture of the software, including the technology stack, the relationship between classes and the Graphical User Interface (GUI) if necessary. In our framework, we ask the Architect agent to write the architecture in Json format, which can be converted to Unified Modeling Language (UML) diagrams using the Mermaid tool (Sveidqvist & Contributors to Mermaid, 2014).

**Project Manager.** This Project Manager agent schedules a code plan based on the architecture diagram. For simplicity, the code plan is represented by a list of files to be created. We also ask the Project Manager agent to match all the requirements in the PRD with the files in the code plan, in order to ensure that all the requirements are considered.

**Programmer.** Unlike GPT-Engineer (Osika., 2023) and ChatDev (Qian et al., 2023) where the agent generates code solely based on the descriptions of requirement, our Programmer agent generates

code based on task plans and architecture diagrams outputted by previous agents. Benefits from appropriate task decomposition and scheduling, our Programmer agent is able to generate code for more complex software development tasks.

**Test Engineer.** Existing works have explored using LLMs for software test from different perspectives, including unit test case generation Li & Yuan (2024) and GUI test Liu et al. (2024). In our framework, due to the diversity of development tasks, the Test Engineer agent focuses on testing general qualities of the code, as is introduced in Section 4.1.3. However, our framework allows different choices of test engineers depending on the characteristics of the task.

### 3.1.1 SHARED CERTIFIED REPOSITORY

Our framework maintains a shared certified repository (SCR) to store intermediate deliverables, including PRD, architecture diagrams, task plans and codes. Each generated content will go through an alignment checking procedure before being added to the SCR, so that to ensure its correctness. The SCR is used in two phases. First, in each content generation phase, agent can flexibly retrieve useful contents from the SCR in order to complete her subtask. Second, during the alignment checking phase, all agents will check the consistency and correctness of the newly generated content based on the SCR. Actually, the SCR represents the agents' common understanding of the task and ensures that the development process runs on the right way.

### 3.2 REAL-TIME ALIGNMENT MECHANISM

It is very common in real-world software development that team members have different understandings of the task, which might significantly impede the development process. Through decades of practices, people have developed many solutions to address the misalignment of understandings. For example, a quick group meeting can help team members to align their opinions when some problems arise. Such a real-time alignment mechanism in teamwork is key to software development since it effectively avoids the failure of the whole project.

In this work, we aim to integrate the Waterfall model with a carefully designed real-time alignment mechanism to accelerate LLM-based multi-agent software development. Our real-time alignment mechanism consists of an alignment checking and a multi-agent discussion phases. The alignment checking phase is compulsorily inserted into the end of each generation phase in the main workflow. The multi-agent discussion phase is optional, conditioned on whether the generated contents pass the alignment checking. We elaborate the two phases in the following sections.

### 3.2.1 ALIGNMENT CHECKING

The goal of alignment checking is to ensure that the newly generated contents correctly realize the requirements and are consistent with the intermediate deliverables in the shared certified repository. As is shown in Figure 2, our framework includes four alignment checking phases. In each alignment checking phase, we call the agent responsible for the current generation phase *initiator*, and all the previous agents *supervisors*. For example, in the third alignment checking phase, the Project Manager serves as the initiator, the Product Manager and the Architect serve as supervisors. Note that the user can optionally participate in the alignment checking, depending on the human resources.

During each alignment checking phase, the initiator sends the newly generated contents to all supervisors. Then, each supervisor retrieves her outcomes from the SCR and checks if the newly generated contents are consistent with her previous outcomes. Note that the supervisors will look into the newly generated contents from different perspectives. For example, in the alignment checking initiated by the Programmer, the Architect who is responsible for generating architecture diagrams would focus on whether the generated codes follow the structural relationships between classes.

However, the alignment checking procedure is non-trivial. If we directly feed the files to an LLM and ask it to check the consistency, it is highly possible that the LLM just outputs a random result. The potential challenges are two-fold. First, the files to be checked usually contain a large amount of information, and the LLM might ignore some important details that determines the checking result. Second, it is hard to develop a unified checking procedure for all of the four alignment checking phases, since different supervisors have different expertise and focuses.

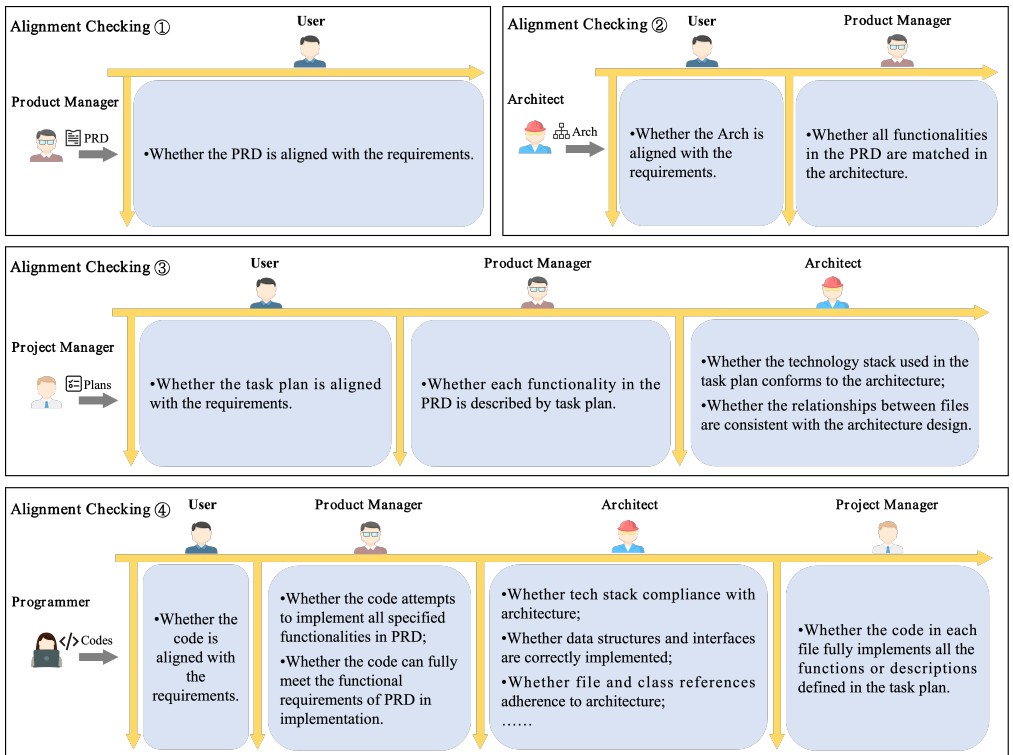

Figure 3: Illustration on the Chain-of-Check prompting in four alignment checking phases. Generally, CoC consists of a primal chain that determines the order of the supervisors, and a subchain that specifies the necessary checking points for each supervisor. Note that the user involvement is optional, depending on whether the human resources are adequate.

To resolve the above challenges, we propose a novel Chain-of-Check (CoC) reasoning procedure to guide the alignment checking in each of the four phases, inspired by the Chain-of-Thought (CoT) (Wei et al., 2022). The key idea of CoC is to decompose each alignment checking procedure into several important steps. As is shown in Figure 3, CoC consists of a primal chain that involves all the supervisors, and a subchain for each supervisor indicating several necessary checking points. The supervisor will approve the current content only when all the checking points in the subchain are satisfied. Overall, a concrete CoC prompt includes the contents to be checked, the contents generated by the supervisor, the necessary checking points and a counter-example explaining the misalignment. More details of the CoC prompts can be found at Appendix A.

The alignment phase ends when all supervisors make decisions on whether to approve the current contents. If all the supervisors approve the newly generated contents, they will be added to the SCR. Otherwise, a multi-agent discussion phase will be initiated to address the misalignments.

### 3.2.2 MULTI-AGENT DISCUSSION

In human development teams, a quick discussion or meeting is perhaps the most efficient way to align with each other. Existing works have shown that discussion in LLM-based multi-agent communities also help to resolve conflicts and accelerate the teamwork Park et al. (2023); An et al. (2024). We introduce an optional multi-agent discussion phase in our framework to efficiently resolve any misalignment identified by the supervisors. Note that in the above alignment checking process, each supervisor checks the initiator's content only from their own perspectives. Since there might be conflicts in the supervisors' reviews, the initiator would be confused when regenerating her contents. Therefore, the multi-agent discussion phase helps to form a more clear feedback to the initiator. Moreover, supervisors in the discussion could refer to others' opinions and revise mistakes they may have made during the alignment checking phase.

Specifically, after the alignment checking, supervisors who vote for misalignment will participate in the multi-agent discussion phase (despite the cases where only one supervisor is involved). The supervisors who vote for alignment do not participate the discussion for the sake of efficiency, but their reviews are visible to the agents in the discussion phase. There are two loops in the multi-agent discussion phase, as is shown in Figure 2. In the inner loop, each agent analyzes other agents' reviews and modifies her own reviews, during with she could flexibly retrieve any necessary documents in the SCR. In the outer loop, the initiator regenerates her contents based on the final reviews outputted by the supervisors after discussion, until the regenerated contents pass the alignment checking or the maximum number of iteration reached. We will show how to determine the maximum numbers of iterations of both inner and outer loop in the next section.

## 4 EXPERIMENTS

### 4.1 EXPERIMENTAL SETTINGS

#### 4.1.1 BASELINES

As LLM-based software development is an emerging topic, we choose three open-sourced representative works as our baselines. GPT-Engineer (Osika., 2023) is a fundamental single-agent approach in LLM-based software agents with a precise understanding of task requirements and the application of one-step reasoning, which highlights its efficiency in generating detailed software solutions at the repository level. MetaGPT (Hong et al., 2024) is an advanced framework that allocates specific roles to various LLM-based software agents and incorporates Standardized Operating Procedures to enable multi-agent participation. In each step, agents with specific roles generate solutions by adhering to static instructions predefined by human experts. ChatDev (Qian et al., 2023) is also a multi-agent software development framework that utilizes a chat chain and communicative de-hallucination to guide specialized agents in collaboratively and autonomously developing software through multi-turn dialogues.

#### 4.1.2 DATASETS

The Software Requirement Description Dataset (SRDD) introduced by Qian et al. (2023) represents a comprehensive corpus of textual software requirements, specifically curated to facilitate agent-driven software development. This dataset includes 1,200 distinct software tasks, which are categorized into five primary domains: Education, Work, Life, Game, and Creation. The dataset incorporates software descriptions from major platforms such as Ubuntu, Google Play, Microsoft Store, and Apple Store, providing a diverse and representative sample of software requirements across various domains. However, we thoroughly reviewed the description of each software requirement and noticed several issues. First, some descriptions were incomplete, ending with ellipses, while many others had similar or even identical functionalities. Second, some tasks are too hard for LLM-based software development at the current stage, such as developing a Monster Hunter game. We believe that these tasks are not suitable for evaluating the automatic software development frameworks. After careful selection, we curated a dataset consists of 214 complete and reasonable tasks.

#### 4.1.3 METRICS

Software testing is a complicated task by its nature. In real-world scenarios, people have developed various approaches for software testing, including unit test Li & Yuan (2024) and GUI test Liu et al. (2024). However, due to the lack of efficient testing tools for large-scale diverse software tasks, existing works usually use simplified metrics to test the quality of generated software Qian et al. (2023); Hong et al. (2024). Following existing works, we choose structural completeness and executability as metrics to evaluate AltDev. In addition, we propose a new functional completeness metric to evaluate how the developed software meets the functional requirements listed in the PRD. The three metrics are elaborated as follows.

• *Structural Completeness* (SC) is a statistical metric that measures the completeness of code structures. In practice, codes generated by LLMs are usually incomplete, where some important classes and functions are replaced by placeholders (e.g., PASS, TODO). For each software development task, we flag it as structural complete if the generated codes do not contain any placeholders, and in-

Table 1: Overall performance of the LLM-based software development methods. Performance metrics are averaged for all tasks. The top scores are in bold, with second-highest underlined.

| Method | Paradigm | SC | Executability | FC |
|---|---|---|---|---|
| GPT-Engineer | single-agent | 0.7009 | 0.8224 | 0.5007 |
| MetaGPT | multi-agent | 0.6822 | 0.6215 | 0.5485 |
| ChatDev | multi-agent | 0.8971 | 0.7383 | 0.5547 |
| AltDev (w/o RTA) | multi-agent | 0.6028 | 0.9018 | 0.5311 |
| AltDev | multi-agent | **0.9205** | **0.9065** | **0.6442** |

complete otherwise. The SC of a set of tasks are calculated as the percentage of structural complete tasks. A higher SC score of a framework indicates the better ability to generate complete code.

• *Executability* measures whether the generated codes could run successfully within the compiling environment environment, regardless of whether the functional requirements are met. In other words, the executability focus on checking if there are low-level bugs in the code. We calculate this metric as the percentage of tasks that compile and run successfully. A higher score indicates the better ability of the framework to generate bug-free code.

• *Functional Completeness* (FC) measures how the software meets the functional requirements. The FC of a task is calculated as the percentage of the accomplished requirements to the total requirements. We report the FC of a development framework as the average FC of all tasks. Compared with the SC and executability metrics, FC is a more advanced metric because it measures the quality of generated code from a higher level.

### 4.1.4 IMPLEMENTATION DETAILS

In practice, calculating accurate FC is hard because it requires a complete test of software functionalities. An alternative approach is to use an LLM for functional test, as is introduced in Tuffveson et al. (2024). In order to verify the LLM's ability for functional test, we conduct an additional set of experiments described as follows. First, we randomly select 25 tasks from the SRDD and generate the codes for them using ChatDev. The 25 tasks correspond to 232 functional requirements in total. Then, we manually examine whether the requirements are accomplished by the generated codes and save human verification results as ground-truth labels. Also, we use GPT-4o to verify if the requirements are accomplished and save the results as predicted labels. The experimental results show that the verification results of GPT-4o achieve an accuracy of 72.84% and a recall of 81.81%. This results support us to use GPT-4o to perform FC evaluation, in order to save human efforts.

In order to reduce human factors in the experimental results, we skip the steps in alignment checking where human inputs are required. Although effective human feedback may help to improve the performance of our methods, we believe that removing human factors leads to a fairer comparison. In addition, we set the maximum number of iterations as 1 for the inner loop, and 3 for the outer loop in the multi-agent discussion phase. We use GPT-4o with a temperature of 0.2. All baselines in the experiments share the same hyper-parameters and settings.

### 4.2 MAIN RESULT

As shown in Table 1, AltDev performs better than all baseline methods in all metrics, which demonstrates the superiority of our framework. In terms of SC, ChatDev achieves the second best result among all methods, benefiting from its code completion phase that involves multiple rounds of code improvement. AltDev achives the best results because our real-time alignment mechanism can effectively detect codes that are not fully implemented or have placeholders. Also the multi-agent discussion greatly improves structural completeness of regenerated codes. In terms of the executability, AltDev also achieves the best results, potentially due to the feedback from the Test Engineer so that the low-level bug can be fixed in real time. FC is the most important metric, which can comprehensively measure whether the code meets the functional requirements. We can see that AltDev has a significant advantage in terms of FC compared with baselines, which indicates that the real-time alignment mechanism indeed ensures that all the agents working towards the same goal.

Table 2: Statistics of tokens (number of tokens used), files (number of code files generated), and lines (total number of lines of codes).

| Method | Tokens | Files | Lines |
|---|---|---|---|
| GPT-Engineer | 2875.28 | 6.67 | 113.15 |
| MetaGPT | 46824.35 | 5.75 | 364.00 |
| ChatDev | 28394.71 | 4.25 | 129.45 |
| AltDev (w/o RTA) | 5355.28 | 3.80 | 133.45 |
| AltDev | 48373.19 | 4.34 | 171.80 |

We also report some statistics of our experiments in Table 2. We can see that although multi-agent approaches (MetaGPT, ChatDev and AltDev) consumes more tokens than the single-agent approach (GPT-Engineer), they also generates larger code bases, which may enhance the integrity and support more functionalities of the software. By analyzing the codes manually, we found that the multi-agent approaches prefer to do more optional works, such as creating a GUI or an interface to a game, even though these additional functionalities are not required. By contrast, GPT-Engineer prefers to be concise in code generation. As a result, it can be found that GPT-Engineer achieves good performance in terms of SC and executability, but performs poor in terms of FC.

AltDev (w/o RTA) is an implementation of AltDev, where the real-time alignment mechanism (including the alignment checking and the multi-agent discussion) is removed. From Table 2 we can find that AltDev consumes more tokens than AltDev (w/o RTA) due to the real-time alignment mechanism. However, by comparing AltDev (w/o RTA) with AltDev in Table 1, we can find that AltDev improves all metrics by a large extent, which demonstrates the effectiveness of the real-time alignment mechanism. In addition, we find that the performance of AltDev (w/o RTA) in terms of executability is very close to that of AltDev, which indicates that the real-time alignment mechanism plays a small role in improving the executability.

### 4.3 GRID SEARCH FOR OPTIMAL HYPER-PARAMETERS

The maximum numbers of inner and outer loop iterations are two key hyper-parameters in Alt-Dev. Increasing the numbers of iterations might lead to better performance, but it also brings more costs. Therefore, we need to find a balance between performance and efficiency. In practice, we randomly sample 20 tasks from the SRDD and run a grid search procedure to find the optimal hyper-parameters. In order to reduce the number of hyper-parameters, we skip the first three alignment checking phases and only run the alignment checking and multi-agent discussion phases initiated by the Programmer agent, as the last alignment checking phase involve all the agents. We set the maximum number of outer loop iterations (A in Figure 4) as 1,2,3,5,7 and the maximum number of inner loop iterations (M in Figure 4) as 1,2,3, resulting in 15 sets of experiments in total.

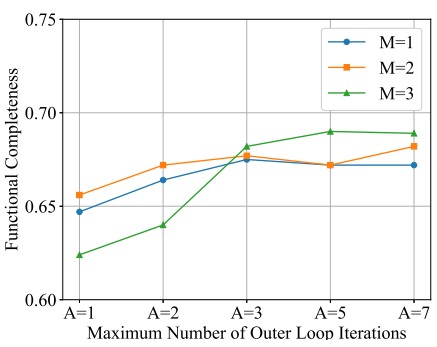

Figure 4: The Functional Completeness performance of AltDev under different hyper-parameter settings in 20 tasks.

As shown in Figure 4, overall, with the increase of the maximum number of outer loop iterations, the FC performance also improves clearly. By averaging over M, we find that the performance on FC increases gradually from A = 1 to A = 7. This also aligns with out intuition that more rounds of regeneration and alignment checking help to improve the quality of code. Note that the number of alignment checking in our work refers to the maximum allowable number of checking, so the outer loop is possibly finished before reaching the limit. This termination often happens for simpler projects, therefore increasing the number of alignment checking is more suitable for addressing complex projects. In our experiments, we choose A=3 since it provides a balance.

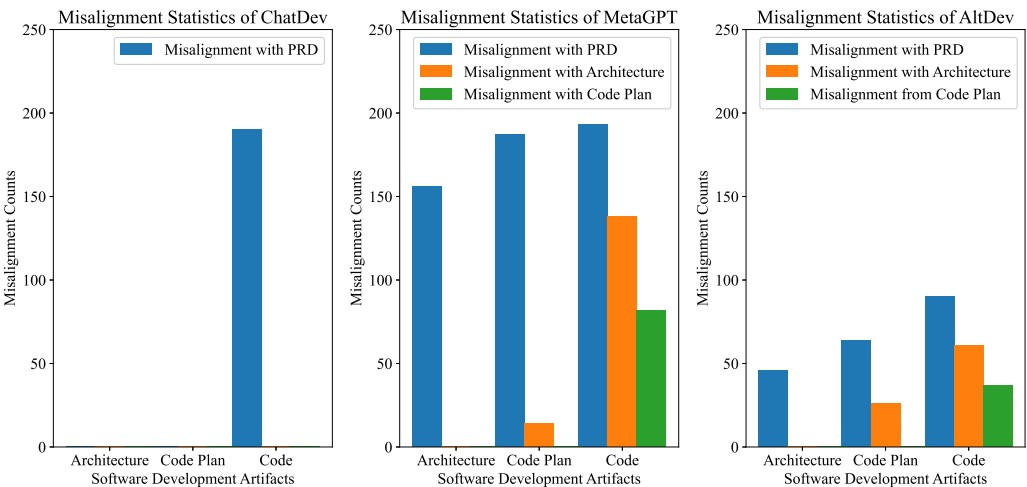

Figure 5: Comparison of misalignment counts.

Regarding to the number of inner loop iterations, the intuition is that more discusses between supervisors could lead to better results. However, as shown in Figure 4, we do not find a clear trend of improvement by increasing M, especially when A is small. As the number of alignment checking increases, the benefits of discussion start to manifest. Overall, the increase on the number of inner loop iterations shows a slight improvement, which is less important than that of the outer loop.

### 4.4 MISALIGNMENT ANALYSIS

Figure 5 shows the misalignment statistics of different frameworks. For MetaGPT and AltDev, we conduct misalignment statistics across three phases in software development: architecture design, code planning and code writing. Considering that ChatDev only generates code, we conduct misalignment statistics during code writing phase and count the misalignment between the PRD and the code. In each phase, we use previous documents relevant to current phase as references to determine whether the contents generated in the current phase are aligned.

Compared to the baselines, AltDev effectively reduces the misalignment at most phases, indicating that by incorporating real-time alignment, AltDev can efficiently resolve misalignment so that to ensure the quality during the whole development process. The only exception is a slight increase (5.6%) in misalignment between architecture and code plan comparing with MetaGPT. This is because that MetaGPT prompts hard encode a strict correspondence between architecture and code plan, which reduces misalignment between these two contents.

## 5 CONCLUSION

In this paper, we introduce AltDev, a novel LLM-based multi-agent framework for software development. AltDev extends the classic Waterfall model by introducing an efficient real-time alignment mechanism. The real-time alignment mechanism includes a compulsory alignment checking phase and an optional multi-agent discussion phase to address misalignment between agents. The experimental results show that AltDev outperforms other baselines, including single-agent approach and multi-agent approaches, in terms of different metrics. Moreover, AltDev significantly performs better in terms of functional completeness, which indicates that AltDev has a great potential in real-world complex software development tasks.

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

# A  APPENDIX

## A.1  CASE STUDY

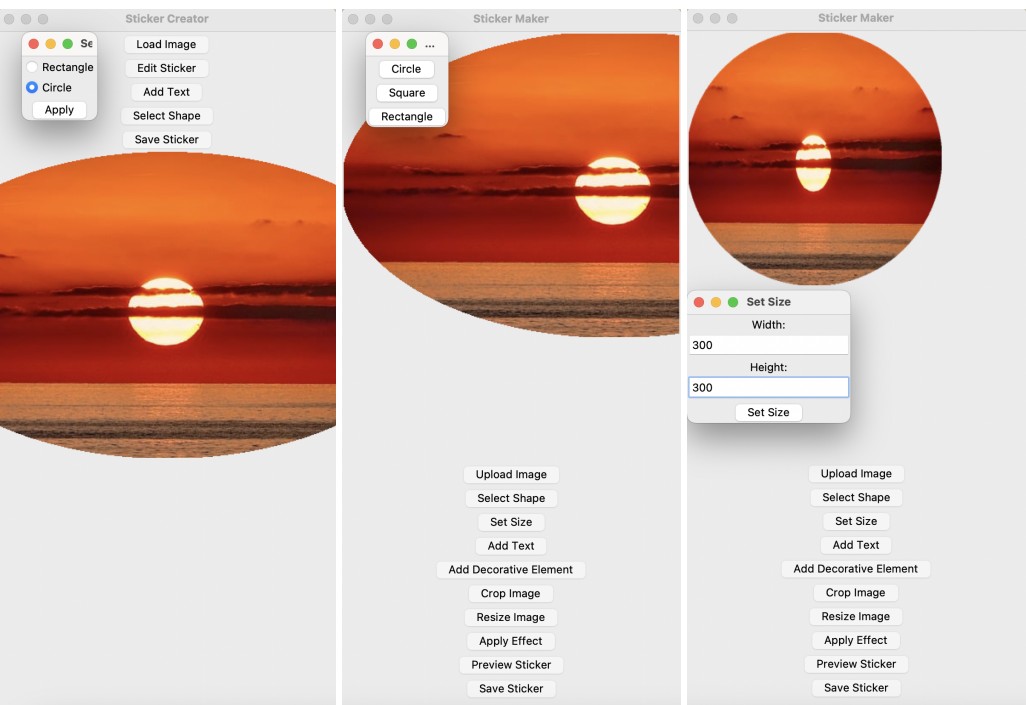

Figure 6: An example of software generated by ChatDev(left) and AltDev(middle and right). In the left picture, we uploaded an image and selected the sticker shape as a circle, and then we cannot change the sticker size. In the middle picture, we uploaded an image and selected the sticker shape as a circle. In the right picture, we set the desired sticker size.

We selected a software in the dataset as an example and compared the effects of the software generated by ChatDev and AltDev, which intuitively demonstrated the superiority of AltDev in software development. The requirement for this software is described as: A software that allows users to create customized stickers using their own photos. Users can select an image, choose the desired shape and size of the sticker, and add text or decorative elements. The software provides easy-to-use tools for cropping, resizing, and adding effects to the photos. Once the sticker is created, users can save it as a transparent PNG file to use in messaging apps or social media platforms.

As illustrated in Figure 6, the software generated by AltDev implements all the functionalities mentioned in the software description, while the software generated by ChatDev does not implement functionalities such as cropping, resizing, and adding special effects. In addition, when implementing the functionality where users can choose the desired shape and size of the sticker, the software generated by ChatDev could only select stickers of a fixed size. These examples show that AltDev can reduce the misalignment in the software development process after the real-time alignment phase, and generate more comprehensive and accurate software that meets the requirements.

## A.2 A CONCREATE EXAMPLE

We provide an example of gradually generating a medical diet planner software project using Altdev. We first show a framework diagram, then demonstrate each phase with corresponding agent's prompt and output.

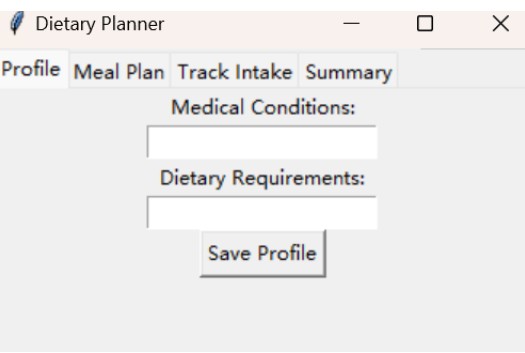

Figure 7: The generated medical diet planner software.

General Workflow

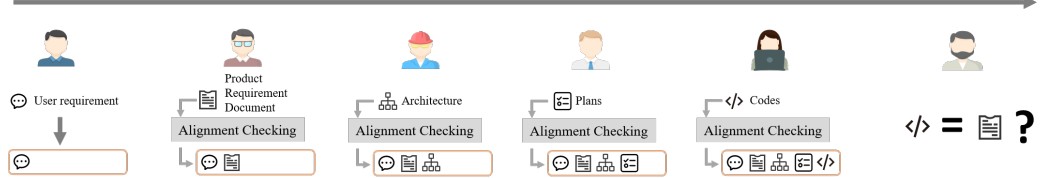

Figure 8: Altdev workflow.

**User**
User needs to input the initial user requirement, typically in natural language.

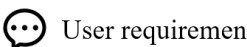 User requirement

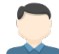 A software application that helps users plan and track their dietary intake based on their medical condition and specific dietary requirements. Provides personalized meal plans, recipe suggestions, and nutritional information. Implemented using Python.

Figure 9: User requirement.

**Product Manager**

Product mananger performs requirement analysis based on the user requirement and generate Product Requirement Document(PRD).

INPUT 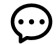  OUTPUT 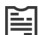

User Requirement  Product Requirement Document(PRD)

**Prompt for generating PRD**

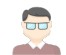

```
role definition

You are an excellent Software Requirements Analyst, and your task is to analyze an
initial project description, generate the functional requirements section of the
Software Requirements Specification (SRS) document (non-functional requirements are
not required). Aim to achieve functional requirements, only require to implement
demo.

# Context
## system description
{user_requirement}
-----
# Format Example
[OUTPUT]
## general overview of the project
## software functional requirements here...
[/OUTPUT]

## Instruction
- general overview of the project: copy the system description here.
- software function requirements: based on the system description, output system's
function requirements

# Constraint
Format: output wrapped inside [OUTPUT][/OUTPUT] like format example, nothing else.

# Action
follow Instruction, Carefully ouput only functional requrirement based on the
original system description.
Aim to achieve functional requirements, only require to implement demo.
do not output ```plaintext or other ``` in the start and the end, output directly.
User Authentication, register, login is not needed.
```

**Product Requirement Document**

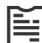

```
## general overview of the project
A software application that helps users plan and track their dietary intake based on
their medical condition and specific dietary requirements. Provides personalized
meal plans, recipe suggestions, and nutritional information. Implemented using
Python.

## software functional requirements
1. The system shall allow users to input their medical conditions and specific
dietary requirements.
2. The system shall generate personalized meal plans based on the user's medical
conditions and dietary requirements.
3. The system shall provide recipe suggestions that align with the user's dietary
needs.
4. The system shall display nutritional information for each meal and recipe
suggestion.
5. The system shall allow users to track their daily dietary intake.
6. The system shall provide a summary of the user's dietary intake over a specified
period.
```

Figure 10: Top: prompt for product manager to generate PRD. Bottom: Generated PRD.

## Architect

Architect analyzes the PRD and determines the general architecture of the software.

INPUT

Product Requirement Document(PRD)

OUTPUT

Architectural diagram

**Prompt for generating Architectural diagram**

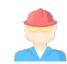

```
role definition

You are a Architect, your goal is design a concise, usable, complete software
system. the constraint is make sure the architecture is simple enough and use
appropriate open source libraries. Aim to achieve functional requirements, only
require to implement demo.
```

```
## Context
functional requirements
{prd}
-----
## format example
[CONTENT]
"Implementation approach": "We will ...",
"File list": ["main.py","game.py"],
"Data structures and interfaces": "
classDiagram
    class Main {
        -SearchEngine search_engine
        +main() str
    }...
    Main --> SearchEngine
    SearchEngine --> Index",
"UI design":"- A canvas for... with ..."
[/CONTENT]

## Nodes: "<node>: <type>  # <instruction>"
- Implementation approach: <class 'str'>  # Analyze the difficult points of the
requirements, select the appropriate open-source framework. If require GUI, you
must also choose a GUI framework (e.g., in Python, you can implement GUI via
tkinter, Pygame, Flexx, PyGUI, etc,)
- File list: typing.List[str]  #  Only need relative paths. ALWAYS write a main.py
here
- Data structures and interfaces: <class 'str'>  # Use mermaid classDiagram code
syntax...
- UI design:<class 'str'>  # ...

## Constraint
Language: Please use the same language as Human INPUT.
Format: output wrapped inside [CONTENT][/CONTENT] like format example, nothing else.

# Attention
1. If a feature of software requires a GUI, you also need to carefully consider...
## Action
Follow instructions of nodes and Attention, generate output and make sure it
follows the format example.
```

Figure 11: prompt for architect to generate architectural diagram.

INPUT     OUTPUT

Product Requirement Document(PRD)     Architectural diagram

**Architectural Diagram**

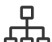

"Implementation approach": "We will use Python as the primary programming language for developing the application. For the GUI, we will use Tkinter, which is a standard GUI toolkit in Python. To handle dietary and nutritional data, we will utilize the 'pandas' library for data manipulation and 'numpy' for numerical operations. For storing user data, we will use SQLite, a lightweight database that is easy to integrate with Python. The 'requests' library will be used for any API calls if needed for fetching additional nutritional data.",
"File list": ["main.py", "dietary_planner.py", "database.py", "ui.py"],
"Data structures and interfaces":

```
class Main {
    -DietaryPlanner planner
    +main() void
}

 class UserProfile {
     -medical_conditions: list
     -dietary_requirements: list
     +update_profile(conditions:
        list,
     requirements: list) void
 }

   class Meal {
      -name: str
      -ingredients: list
      -nutritional_info: dict
      +calculate_nutritional_info()
dict
    }

class Database {
      +save_user_data(user_profile:
        UserProfile) void
      +load_user_data() UserProfile
    }

      Main --> DietaryPlanner
      DietaryPlanner --> UserProfile
      DietaryPlanner --> MealPlan
      MealPlan --> Meal
      Main --> UI
      UI --> DietaryPlanner
      DietaryPlanner --> Database
```

```
class MealPlan {
      -meals: list
      +add_meal(meal: Meal) void
      +get_nutritional_info() dict
    }

class DietaryPlanner {
      -UserProfile user_profile
      -MealPlan meal_plan
      +input_medical_conditions(conditions:
        list) void
      +generate_meal_plan() MealPlan
      +get_recipe_suggestions() list
      +track_daily_intake(meal: Meal) void
      +get_summary(period: str) dict
    }

class UI {
      -DietaryPlanner planner
      +start() void
      +display_meal_plan(meal_plan: MealPlan)
void
      +display_recipe_suggestions(recipes:
list) void
      +display_nutritional_info(info: dict)
void
      +display_summary(summary: dict) void
    }
```

"UI design": "- Main Window: A window with tabs for 'Profile', 'Meal Plan', 'Track Intake', and 'Summary'. Each tab will have specific input fields and display areas.
\n- Profile Tab: Input fields for medical conditions and dietary requirements.\n- Meal Plan Tab: Display area for generated meal plans and nutritional information.\n- Track Intake Tab: Input fields to log daily meals and display current intake.\n- Summary Tab: Display area for dietary intake summary over a specified period."

Figure 12: Generated architectural diagram.

## Project Manager

Project Manager schedules a code plan based on the PRD and the architectural diagram.

---

**INPUT**     Product Requirement Document(PRD)     Architectural Diagram     **OUTPUT**     Code Plan

---

**Prompt for generating Code Plan**

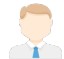

```
role definition

You are a Project Manager, named Eve, your goal is break down tasks according to
functional requirement/architecture, generate a task plan, and analyze task
dependencies to start with the prerequisite modules. the constraint is use same
language as user requirement.
```

```
## Context
functional requirement:
{prd}
architecture:
{architecture}

## Nodes: "<node>: <type>  # <instruction>"
- Required packages: typing.List[str]  # Provide required packages in requirements.txt
format.
- Required Other language third-party packages: typing.List[str]  # List down the
required packages...- Logic Analysis: typing.List[typing.List[str]]  # Provide a list
of files with the classes/methods/functions to be implemented, including dependency
analysis and imports.
- Task list: typing.List[str]  # Break down the tasks into a list of filenames...
- Full API spec: <class 'str'>  # Describe all APIs using OpenAPI 3.0 spec that may ...
- Shared Knowledge: <class 'str'>  # Detail any shared knowledge, like common utility
functions...

## Constraint
Language: Please use the same language as Human INPUT.
Format: output wrapped inside [CONTENT][/CONTENT] like format example, nothing else.

## Action
Follow instructions of nodes, generate output and make sure it follows the format
example.
```

**Code Plan**

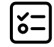

```
"Required packages": [
    "pandas==1.5.3",
    "numpy==1.23.5",
    "tkinter",  # Tkinter is included with Python, no need for version
    "sqlite3",  # sqlite3 is included with Python, no need for version],
"Required Other language third-party packages": ["No third-party dependencies
required"],
"Logic Analysis": [
    ["main.py", "Contains the main function to initialize the application and create
instances of UI and DietaryPlanner.],
    ["dietary_planner.py", "Contains DietaryPlanner class with methods for
input_medical_conditions, generate_meal_plan, get_recipe_suggestions,
track_daily_intake, and get_summary."],
    ["database.py", "Contains Database class with methods for save_user_data and
load_user_data. "],
    ["ui.py"...],
"Task list": ["database.py","dietary_planner.py","ui.py","main.py"],
"Full API spec": "",
"Shared Knowledge": "`dietary_planner.py` and `database.py` contain core logic and data
handling functions shared across the project."
```

Figure 13: Top: prompt for project manager to generate code plan. Bottom: Generated code plan.

**Programmer**

Programmer generates code file based on the architectural diagram and code plan.

INPUT     Architectural Diagram     Code Plan     OUTPUT     Code

**Prompt for generating Code**

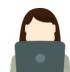

```
role definition

You are a professional engineer; the main goal is to write google-style, elegant,
modular, easy to read and maintain code.
Output format carefully referenced "Format example".
```

```
# Context

## Design
{architecture}
## Task Plan
{task_plan}
-----
# Format Example
### main.py
```python
...
```

### ui.py
```python
...
```

-----
# Instruction: Based on the context, follow "Format example", write code. .

# ATTENTION
1. Use '###' to SPLIT CODE SECTIONS, neither '#' and not '##'. do not forget ``` in
each file, refer the the example. Output format carefully referenced "Format example".
2. Follow design: YOU MUST FOLLOW "Data structures and interfaces". DONT CHANGE ANY
DESIGN. Do not use public member functions that do not exist in your design.
3. Follow task: YOU MUST write Comprehensive codes to complete task of each file in
task list.
4. CAREFULLY CHECK THAT YOU DONT MISS ANY NECESSARY CLASS/FUNCTION IN THIS FILE.
5. You must import the third-party libraries used in your code
6. Determine the order of writing the files based on your understanding of the project.
7. Write out EVERY CODE DETAIL, DON'T LEAVE TODO,PASS,PLACEHOLDER.
8. Only write code result, do not output any other content in the start or in the end.
```

Figure 14: Prompt for programmer to generate code.

INPUT     Architectural Diagram     Code Plan     OUTPUT     Code

**Code**

```python
### database.py
import sqlite3
from dietary_planner import UserProfile

class Database:
    def __init__(self, db_name='user_data.db'):
        self.connection = sqlite3.connect(db_name)
        self.cursor = self.connection.cursor()
        self._create_tables()

    def _create_tables(self):
        self.cursor.execute('''
            CREATE TABLE IF NOT EXISTS user_profile (
                id INTEGER PRIMARY KEY,
                medical_conditions TEXT,
                dietary_requirements TEXT
            )
        ''')
        self.connection.commit()

    def save_user_data(self, user_profile: UserProfile):
        medical_conditions = ','.join(user_profile.medical_conditions)
        dietary_requirements = ','.join(user_profile.dietary_requirements)
        self.cursor.execute('''
            INSERT INTO user_profile (medical_conditions, dietary_requirements)
            VALUES (?, ?)
        ''', (medical_conditions, dietary_requirements))
        self.connection.commit()
```

```python
### dietary_planner.py
import pandas as pd
import numpy as np
from database import Database
...
```

```python
### ui.py
import tkinter as tk
from tkinter import ttk
from dietary_planner import DietaryPlanner, MealPlan

class UI:
    def __init__(self):
        self.planner = DietaryPlanner()
        self.root = tk.Tk()
        self.root.title("Dietary Planner")
        self._setup_ui()

    def _setup_ui(self):
        self.notebook = ttk.Notebook(self.root)
        self.notebook.pack(expand=True, fill='both')

        self.profile_tab = ttk.Frame(self.notebook)
        self.meal_plan_tab = ttk.Frame(self.notebook)
        self.track_intake_tab = ttk.Frame(self.notebook)
        self.summary_tab = ttk.Frame(self.notebook)
```

...

Figure 15: Generated code.

**Tester**

Tester test the project and provide feedback to programmer. If error occurs, programmer needs to fix the error and revise codes.

---

INPUT    `</>`    Code        OUTPUT    [ **Test Report** ]

---

**Test Report**

```
Error: Traceback (most recent call last):

  File " altedev\project_dir\Medical_Diet_Planner\review_code\main.py", line 1, in <module>
    from ui import UI
  File "altedev\project_dir\Medical_Diet_Planner\review_code\ui.py", line 3, in <module>
    from dietary_planner import DietaryPlanner, MealPlan
  File altedev\project_dir\Medical_Diet_Planner\review_code\dietary_planner.py", line 3, in
<module>
    from database import Database

  File " altedev\project_dir\Medical_Diet_Planner\review_code\database.py", line 2, in
<module>
    from dietary_planner import UserProfile

ImportError: cannot import name 'UserProfile' from partially initialized module
'dietary_planner' (most likely due to a circular import)
```

---

INPUT    `</>`    Code    [ **Test Report** ]        OUTPUT    `</>`    Code

---

```
### database.py
```python
import sqlite3
from user_profile import UserProfile

class Database:
    def __init__(self, db_name='user_data.db'):
        self.connection = sqlite3.connect(db_name)
        self.cursor = self.connection.cursor()
        self._create_tables()

### dietary_planner.py
```python
import pandas as pd
import numpy as np
from database import Database
from user_profile import UserProfile

class Meal:
    def __init__(self, name, ingredients, nutritional_info=None):
        self.name = name
        self.ingredients = ingredients
        self.nutritional_info = nutritional_info or {}

    def calculate_nutritional_info(self):
        self.nutritional_info = {
            'calories': sum(ingredient['calories'] for ingredient in self.ingredients),
            'protein': sum(ingredient['protein'] for ingredient in self.ingredients),
            'carbs': sum(ingredient['carbs'] for ingredient in self.ingredients),
            'fats': sum(ingredient['fats'] for ingredient in self.ingredients)
        }
        return self.nutritional_info
...
```

Figure 16: Top: Test report. Bottom: The code revised according to the test report.

## A.3 STANDARDIZED PROMPT SEQUENCES

SUPERVISOR
Product Manager

INITIATOR
Architect

Architecture

Alignment Checking

```
You are a Product Manager.
This is a Requirement Document:
{PRD}

This is a Architecture:
{Architecture}
-----
-----
Example:
## example for not match
Requirement Document:
---
2.3. The system shall allow users to choose the shape of the sticker (e.g.,
circle, square, custom shape).

Architecture:
class ImageEditor {
        +upload_image(file_path: str) Image
        +add_decorative_elements(image: Image, element: str, position: tuple)
Image
        +save_image(image: Image, file_path: str) void
    }
# Not match. The architecture does not explicitly mention the function of
selecting shapes. need to add relevant methods in the ImageEditor class and add
a shape selection menu in the GUI class.
---
......

final Summary: [NOTMATCH]

## example for match
Requirement Document:
---
2.5. The system shall allow users to add text to the sticker.

Architecture:
+add_text(image: Image, text: str, position: tuple, font: str, size: int, color:
str) Image
# match. add_text() mention requirement of add text to the sticker.
---
....

**final summary: [MATCH]**

------

# Action
Analyze whether all the functions in the requirements are match in the
architecture(such as Class and Function).
Add a summary for each analysis, whether it is match or not. use --- to
separate each requirement check.
In the final summary, output whether it is MATCH or NOTMATCH(warpped in [],
[MATCH] for MATCH summary and [NOTMATCH] for NOTMATCH summary).
Only output [MATCH] or [NOTMATCH] in final summary based on your analysis.
Follow Example and output your result.
```

Figure 17: Prompt of alignment checking between PRD and architectural diagram.

SUPERVISOR

Product Manager

INITIATOR

Project Manager

Code Plan

Alignment Checking

```
You are a Product Manager.
This is a Requirement Document:
{prd}
This is a Code Plan:
{code_plan}
------
# Example
## example for not match
---
The system shall allow users to create 3 tools include pencil, brush and spray
gun.
    ...
    [
        "brush.py",
        "Contains various brushes to let user select."
    ]
    ...
# Not match. Requirement to create three types of brushes lost, need to point
out the various brush types include pencil, brush, spary gun.
---
......

**final Summary: [NOTMATCH]**

## example match
---
The system shall allow users to create 3 tools include pencil, brush and spray
gun.
    ...
    [
        "brush.py",
        "Contains 3 types of brushes(pencil, brush, spary gun) to let user
select."
    ]
    ...
# match.
---
......

**final Summary: [MATCH]**
------

# Instruction
(1)Each file name in the Logic Analysis is followed by the description that the
file is responsible for. These files do not have code yet, so you only need to
judge from these functional descriptions.
(2)File description in the Task List must accurately and completely match the
requirement it is responsible for. Otherwise it's summary is NOTMATCH.

# Action
Carefully analyze whether each feature in the requirements is correctly and
accurately described by Logic Analysis in task plan.
Add a summary after each analysis, whether it is match or not. use --- to
separate each requirement check.
In the final summary, output whether it is MATCH or NOTMATCH(warpped in [],
output [MATCH] for MATCH summary and [NOTMATCH] for NOTMATCH summary).
```

Figure 18: Prompt of alignment checking between PRD and code plan.

SUPERVISOR

INITIATOR

Architect

Project Manager

Code Plan

Alignment Checking

```
You are a An software Architect.
# Architecture
{architecture}
# Code Plan
{code_plan}
-----
# example for match case
## Architecture
class StickerCreator {{
        ...
        +resize_photo(new_size: tuple) None
        +apply_effect(effect: str) None
        ...
    }}

## Task Plan

"sticker_creator.py",
"Contains StickerCreator class with methods for uploading photos, selecting
shapes, setting sizes, adding text and decorative elements, rezize and
apply_effect...

# match, sticker_creator.py in task plan contains the appropriate classes and
functions in architecture...
---
final summary: [MATCH]

# example for not match case
## Architecture
...
---
class StickerCreator {{
        ...
        +resize_photo(new_size: tuple) None
        +apply_effect(effect: str) None
        ...
    }}

## Task Plan
...
"sticker_creator.py",
"Contains StickerCreator class with methods for uploading photos, selecting
shapes, setting sizes, adding text and decorative elements, cropping,...

# Not match, sticker_creator.py in task plan omit the function of apply_effect
in architecture
---
final summary: [NOTMATCH]

# Action
Carefully analyze whether the technology stack used in the task Plan conforms
to the architecture, and whether the relationships between files are consistent
with the architecture design.
In the final summary, output whether it is MATCH or NOTMATCH(warpped in [],
output [MATCH] for MATCH summary and [NOTMATCH] for NOTMATCH summary).
```

Figure 19: Prompt of alignment checking between architectural diagram and code plan.

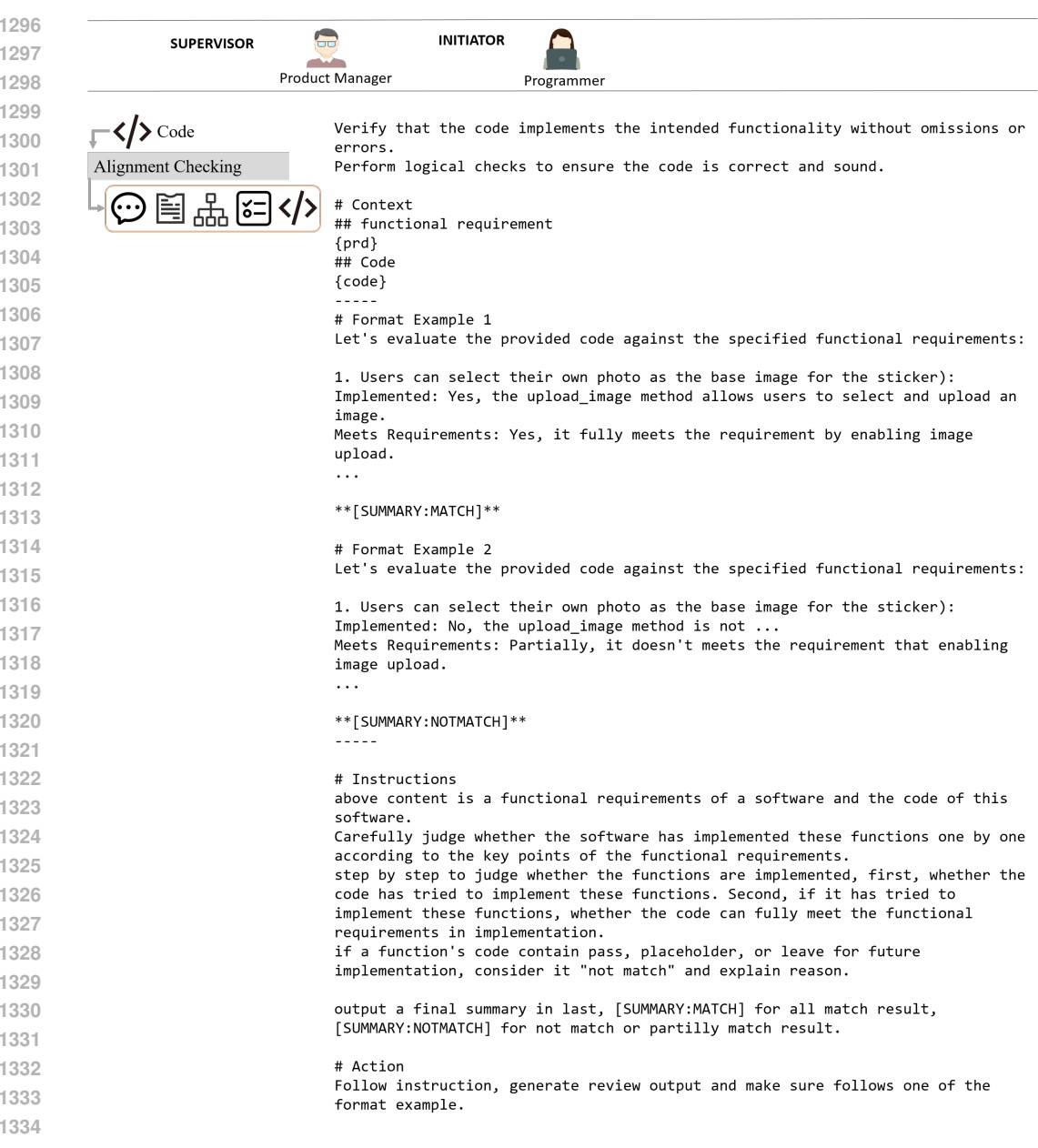

Figure 20: Prompt of alignment checking between PRD and code.

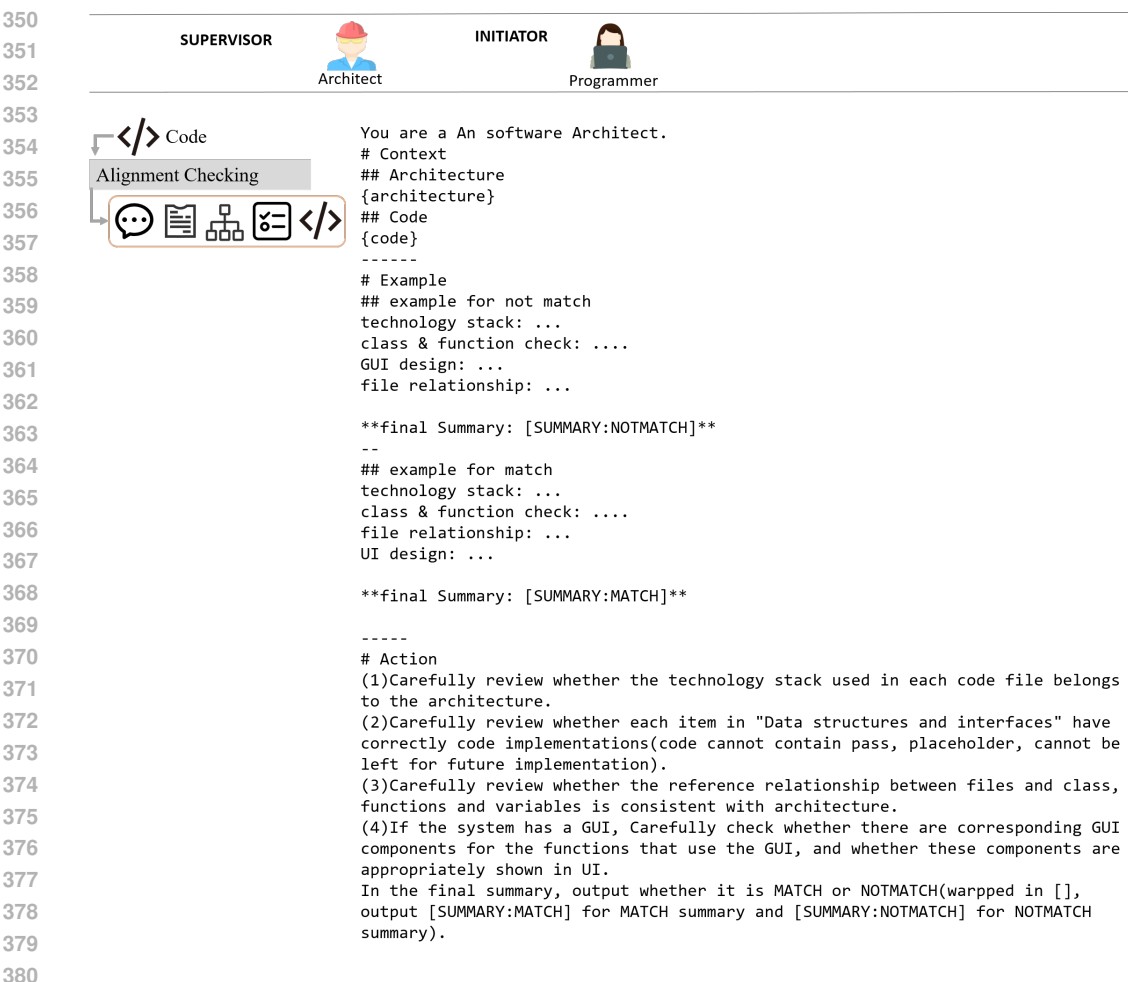

Figure 21: Prompt of alignment checking between architectural diagram and code.

SUPERVISOR          INITIATOR

Project Manager          Programmer

</> Code

Alignment Checking

```
You are a An software Project Manager.
# Context
# Code Plan
{code_plan}
## Code
{code}

# Example
## example for not match
---
Task:
"health_profile.py",
"Contains HealthProfileManager class with methods to create, update, and
retrieve health profiles."
Code:
Implemented: no, the create method in healthy_profile.py is not implement by
code.
Implemented: no, the create method need a button to trigger but no code
implementation.
---
...

**final Summary: [SUMMARY:NOTMATCH]**
---
...

## example for match
---
Task:
"health_profile.py",
"Contains HealthProfileManager class with methods to create, update, and
retrieve health profiles."
Code:
Implemented: yes, the create method in healthy_profile.py is implement by code,
user can input their profile.
(optional if required ) GUI Implemented: Yes, the main UI has a button to
trigger create function.
---
...

**final Summary: [SUMMARY:MATCH]**

-----
# Action
step by step, Carefully analyze whether each file in the code contains code
that fully implements the all functionality or description defined by the file
with the same name in the Logic Analysis in Task Plan.
output [SUMMARY:NOTMATCH] because code can not contain pass, placeholder, can
not be left for future implementation.
In the final summary, summary previous result and output whether MATCH or
NOTMATCH(warpped in []), output [SUMMARY:MATCH] for MATCH summary and
[SUMMARY:NOTMATCH] for NOTMATCH summary).
```

Figure 22: Prompt of alignment checking between code plan and code.

## A.4  MULTI AGENT DISCUSSION

We provide a multi-agent discussion prompt during the code alignment process as an example.

---

**Multi Agent Discussion**

---

**Prompt for Multi Agent Discussion**

```
You are {role}, You are reviewing a Code based on your content.
You have generated your review result, and others have also generated
review result, All of you are in a team.

---
# Context
## {role_own_content}
{roles_own_content} # such as PRD, architecture, plan, for
corresponding roles.

## Code
{code}

# Review Result
## Your result
{role_alignment_checking_result}
## other's review result
{others_alignment_checking_result}

# Action
First, you need to carefully analyze other's review result of Code and
your review result.
Second, Using other's review result as reference, based on your
functional requirement document, you need to regenerate a new review
result of the Code.

# Constraint
(1)format of Regenerated result must carefully follow your original
review result' format.
(2)no need to copy others' opinions directly.
(2)Do not need to explain in the start and end.
```

Figure 23: Prompt for Multi-Agent Discussion.

