# OpenReview forum: "AltDev: Achieving Real-Time Alignment in Multi-Agent Software Development"
_ICLR.cc/2025/Conference — ICLR 2025 Conference Withdrawn Submission_

### Official Review · Reviewer_2RBF · 2024-11-01

**Soundness:** 2
**Presentation:** 2
**Contribution:** 2
**Rating:** 3
**Confidence:** 4

**Summary:**

This paper introduces AltDev, a multi-agent framework for software development that addresses the misalignment issues in LLM-based collaborative coding. Unlike existing frameworks that follow linear development models, AltDev implements a real-time alignment mechanism that enables agents to correct their deliverables and align with other agents throughout the development process. The framework integrates an alignment checking and a conditional multi-agent discussion at each development phase, allowing early identification and reduction of errors. Through experiments on software development tasks, the authors demonstrate that AltDev significantly improves code quality in terms of executability, structural completeness, and functional completeness compared to existing approaches.

**Strengths:**

1. The paper addresses a significant challenge in multi-agent software development through a well-designed pipeline that incorporates real-time alignment checking and multi-agent discussion phases to catch and correct errors early.

2. The experimental results are good, showing improvements across multiple metrics compared to baseline approaches.

**Weaknesses:**

1. Potential violation of double-blind review policy: The paper provides a non-anonymized Github link in the abstract. This could potentially violate the anonymity requirement of ICLR submission.

2. Many crucial implementation details are vaguely described:
   - The mechanism for agents to retrieve and verify contents from SCR is unclear
   - The multi-agent discussion process lacks details about the communication, e.g., communication order and information access.
   - The "Chain-of-Check" reasoning procedure, while presented as a key contribution, lacks detailed explanation of its implementation, e.g., who proposes the checking points for each supervisor?

3. Scalability concerns are not properly addressed:
   - As development progresses, the number of agents involved in alignment checking increases, because all the previous agents should participate the alignment checking, which could lead to significant overhead and efficiency issues in larger projects
   - The token usage statistics in Table 2 are ambiguous - it's unclear whether it is inference tokens alone or include prompt tokens

4. Lack of concrete evidence and case studies. The benefits of RTA mechanism are claimed but not demonstrated through specific examples. Readers can better understand the effectiveness of RTA from examples with and without RTA.

5. The evaluation is done only on one dataset, which may not be sufficient

Overall, while the paper presents an interesting approach to multi-agent software development, the lack of crucial implementation details, scalability concerns, and insufficient case studies make it difficult to fully assess the method's practical value.

**Questions:**

See weaknesses above

**Details Of Ethics Concerns:**

Potentially violates the double-blind review policy by providing non-anonymized Github link in the abstract.

---

### Official Review · Reviewer_uVpA · 2024-11-03

**Soundness:** 3
**Presentation:** 4
**Contribution:** 3
**Rating:** 6
**Confidence:** 2

**Summary:**

The paper develops a multi-agent system for real time code generation on software developing tasks. The framework is based on agents playing different roles of software development, allowing for real-time alignment between the different agents, in order to improve generated code quality. The authors compare their proposed system to existing baselines and find an improvement in performance quantified by several evaluation metrics.

**Strengths:**

- Topic of interest - In complex coding tasks, contemporary LLMs tend to struggle to achieve good performance. Improving upon this limitation is of great interest.
- Interesting approach - The paper approaches this problem with a multi-agent system, that mimics human software teams.
- Results indicate improvement in performance - combination of high structural correctness + executability.
- While previous works have used multi-agent systems for code generation tasks (see related works below), to the best of my knowledge, the paper still presents a novel implementation of a social simulation for large scale code development. The combination of using a role-paying multi-agent system with the task of creating complex softwares is interesting.

**Weaknesses:**

- Functional correctness is a crucial aspect of evaluation. If I understand correctly, the accuracy of GPT4-o, used to determine functional correctness is not very high (73% precision, 82% recall). If this is the case, it weakens the statistical significance of the results displayed in table 1 on FC - while AltDev with RTA does indeed have significantly higher FC than the other methods (~20%), which does point to a significance (as the others have relatively concentrated lower FCs), it is not super assuring, as the improvement of AltDev is on the same scale as the "testing unit"'s accuracy. Is it possible to analyze the sensitivity of the results to GPT-4o's accuracy?
- Related works - while the method proposed in this work is novel in the type of multi-agent interaction that is implemented, I would mention the works [1,2] that deal with coding and mathematical problems and explain the difference to their multi-agent systems, in order to provide a more complete picture on the state of this research field (those works are of course different in the type of problem they try to solve and implementation, and this difference can be elaborated on). I also believe the specific type of multi-agent interaction proposed in this work resembles the one used in [3], in which social simulations with roleplay were used for self alignment of LLMs (the target problem is very different, but the idea of multi agent role play is similar), so it would be appropriate to mention.
- (Minor) - writing style can be improved (typos).
- All that being said, I appreciate the approach and am open for a fruitful discussion.

[1] - https://arxiv.org/abs/2404.02183

[2] - https://arxiv.org/abs/2310.02170

[3] - https://arxiv.org/abs/2402.05699

**Questions:**

- Am I missing something about the functional correctness? Perhaps a simple explanation about why GPT4-o's accuracy is good enough to determine the FC.

---

### Official Review · Reviewer_FbxW · 2024-11-04

**Soundness:** 2
**Presentation:** 3
**Contribution:** 2
**Rating:** 3
**Confidence:** 4

**Summary:**

This paper presents a multi-agent framework aimed at addressing the issue of uncoordinated communication among multiple roles in software engineering development. Traditional waterfall models merely pass development requirement information sequentially from one role to another, which can result in cumulative errors due to misinformation at any stage. To tackle this, the paper proposes an AltDev multi-agent framework that incorporates a multi-role collaborative loop check at each stage of the information transfer process to help ensure the accuracy of information transmission. The framework is ultimately evaluated on the SRDD test set, where it is compared with other multi-agent solutions (such as Meta-GPT and ChatDev). The final experimental results indicate that the proposed AltDev framework demonstrates improvements in both completeness and executability metrics.

**Strengths:**

1. This paper addresses the issue of communication misalignment among multiple roles in software engineering development and proposes a multi-agent framework to solve this problem.
2. The paper demonstrates through experiments that AltDev achieves higher completeness and executability on the SRDD test set compared to conventional multi-agent frameworks.

**Weaknesses:**

1. There is no comparison between the proposed multi-round, multi-role, collaborative check approach and the ordinary rethinking method used by a single role to determine if there is a significant improvement. In other words, a stronger baseline should be established, such as adding some simple rethinking mechanisms to chatdev.
2. Can simply incorporating user requirements into the context of each role to some extent mitigate the accumulation of communication errors among multiple roles? This question relates to whether the motivation of the paper is strong.
3. There is a lack of more case studies and detailed analyses to demonstrate whether the collaborative cyclic check among multiple roles truly brings about a significant enhancement in capability.

**Questions:**

1. In Section 4.1.1, more baseline models from the multi-agent domain should be introduced to ensure a fairer comparison, such as XAgent, AutoGen, etc.
2. In Appendix 1.1, more case studies should be included to help determine whether multi-role collaboration issues are common.

---

### Official Review · Reviewer_DxzS · 2024-11-04

**Soundness:** 3
**Presentation:** 2
**Contribution:** 2
**Rating:** 5
**Confidence:** 4

**Summary:**

The paper introduces AltDev, a framework which wraps LLMs to provide a multi-agent system wherein different roles collaborate to design and build software.

The workflow is similar to the waterfall model, with the addition of 'Alignment Checking': stage-gates where roles which have been involved up to that point are asked to check that the most recent product matches that agent's assigned goal. Additionally, the paper also explores the addition of a "real-time alignment mechanism": an optional inter-agent discussion phase which can be voted for by the agents after each generation phase.

Experimentally, the paper compares AltDev's performance on an (unspecified) subset of 214 tasks from the 1,200 tasks in SRDD, to the performance of three baseline agents: GPT-Engineer, MetaGPT and ChatDev.

The metrics used are "Structural Completeness" (absence of placeholder sections), Executability, and "Functional Completeness". FC is a measure of the fraction of requirements which the code accomplishes, and is determined via static analysis using GPT-4o, found to have ~70% agreement with human judgement.

According to these metrics, AltDev outperforms the three competitor agents.

**Strengths:**

The paper correctly identifies that current LLM agents excel at simple coding tasks far more than on complex software-engineering tasks which require long-term planning.

The focus on adaptability and responsivity in plans is a good one.

Removing the requirement for human input during the alignment checking makes the comparison fairer.

Performing an ablation of the RTA component measures the contribution of this specific component.

Comparing to other frameworks provides a baseline which highlights the improved performance of the AltDev technique.

The rounds of agent discussion, with defined roles, may create opportunities for the system to find and resolve bugs.

The case study in Appendix A.1 shows that AltDev can generate functional software from a PRD. Currently the software created is basic, but these early steps are significant.

The concrete example in A.2 is useful for demonstrating the level of complexity achievable by this approach.

**Weaknesses:**

Often poor presentation, including overly-informal tone ("the internal logics of functionalities could be super complicated"), many typos / errors (including two in the first paragraph of the introduction: "However, single LLM performs poor", "promosing"), and vague assertions ("If we directly feed the files to an LLM and ask it to check the consistency, it is highly possible that the LLM just outputs a random result").

While other agents are used as baselines (e.g. GPT-Engineer, MetaGPT), these are different to the AltDev framework in many ways, and so it is hard to tell which components of the framework are responsible for what level of improvement. Real Time Alignment (RTA) is ablated, but a more full-bodied paper would also ablate each of the various contributions. E.g. how important is the role of architect? How important is the role of Product Manager, or Project Manager? Kapoor et al https://arxiv.org/abs/2407.01502v1 show that adding complexity to a framework can vastly increase the cost, with minimal gains.

Minimal details are given on the chosen subset of SRDD.

Minor point: In Fig 5 the three words of the "Software Development Artifacts" title unfortunately lines up with the three labels along the axis, decreasing readability (e.g. "Architecture Software" and "Code Artifacts"). Consider adjusting spacing or using a different font for the two.

**Questions:**

I'm not sure what is meant by the text in parentheses ~L325: "Specifically, after the alignment checking, supervisors who vote for misalignment will participate in the multi-agent discussion phase (despite the cases where only one supervisor is involved)." Does this mean that even when only one agent votes for misalignment, then all agents discuss? Or does it mean that in that case that only one agent votes for misalignment, that they have a discussion with themselves? How would such a discussion proceed?

Which tasks from SRDD were selected, to form the subset of 214 "complete and reasonable" tasks?

When determining Structural Completeness, how are placeholders identified?

GPT-4o is used to determine Functional Completeness. In §4.1.4, the authors compare GPT-4o's results to human verification, with an accuracy of ~70%. How does this accuracy translate to error bars on results? In particular, how do the size of the error bars compare to the differentiation between model performances in Table 1?

According to Table 1, AltDev is still producing Structurally Incomplete code 8% of the time (e.g. code which contains placeholders). Why is this not being caught by the alignment procedures? Same question goes for Executability (10% of the time AltDev misses low-level bugs, generating code which does not compile).

Around L443 it's stated that GPT-Engineer, being a single-agent system, is more concise, doing less optional and unrequired work. Why would this lower its FC score? If efficiency reduces FC, does this make FC a less suitable metric?

Why, when being more efficient during the hyper-parameter search, is it the Programmer agent's initiations which trigger alignment checking? What would happen if a different agent was chosen? How does performance of this more efficient scheme compare to that of the full scheme?

---

### Note · Authors · 2024-11-23

**Comment:**

Thank you for your valuable time and insightful feedback on our manuscript. Your comments have provided us with a deeper understanding of the shortcomings in our work, for which we are truly grateful. After careful consideration of the necessary revisions and the current state of our research, we believe that the manuscript in its present form may not fully meet the standards of your esteemed conference. Therefore, we have decided to withdraw our submission and will focus on further improving the work.

We sincerely appreciate your effort and expertise in reviewing our manuscript and look forward to the opportunity to submit a more refined version in the future.

**Withdrawal Confirmation:**

I have read and agree with the venue's withdrawal policy on behalf of myself and my co-authors.